# Minimizing the Global Warming Potential with Geopolymer-Based Insulation Material with Miscanthus Fiber

**DOI:** 10.3390/polym14153191

**Published:** 2022-08-05

**Authors:** Steffen Witzleben

**Affiliations:** Institute of Technology, Resource and Energy Efficient Engineering, Bonn-Rhein-Sieg University of Applied Sciences, von-Liebig-Strasse 20, 53359 Rheinbach, Germany; steffen.witzleben@h-brs.de

**Keywords:** geopolymer, thermal insulation materials, Miscanthus, fiber composites

## Abstract

Approximately 45% of global greenhouse gas emissions are caused by the construction and use of buildings. Thermal insulation of buildings in the current context of climate change is a well-known strategy to improve the energy efficiency of buildings. The development of renewable insulation material can overcome the drawbacks of widely used insulation systems based on polystyrene or mineral wool. This study analyzes the sustainability and thermal conductivity of new insulation materials made of Miscanthus x giganteus fibers, foaming agents, and alkali-activated fly ash binder. Life cycle assessments (LCA) are necessary to perform benchmarking of environmental impacts of new formulations of geopolymer-based insulation materials. The global warming potential (GWP) of the product is primarily determined by the main binder component sodium silicate. Sodium silicate’s CO_2_ emissions depend on local production, transportation, and energy consumption. The results, which have been published during recent years, vary in a wide range from 0.3 kg to 3.3 kg CO_2_-eq. kg^−1^. The overall GWP of the insulation system based on Miscanthus fibers, with properties according to current thermal insulation regulations, reaches up to 95% savings of CO_2_ emissions compared to conventional systems. Carbon neutrality can be achieved through formulations containing raw materials with carbon dioxide emissions and renewable materials with negative GWP, thus balancing CO_2_ emissions.

## 1. Introduction

The construction of infrastructure and buildings is related to about half of the raw materials and energy consumption in the European Union. Additionally, about one-third of water consumption and waste material produced in Europe is due to construction. The energy used for heating and climatization of living rooms has a huge impact on CO_2_ emissions. Worldwide, about 45% of global greenhouse gas emissions and raw material consumption are related to buildings [1]. In Germany, 14% of CO_2_ emissions can be attributed to buildings and 35.5% to energy production [2]. Due to current energy-saving regulations, thermal insulation systems of houses are widely used and based mainly on mineral wool or polystyrene [3].

The development of insulation materials for houses through the utilization of renewable resources and waste products is a promising route for CO_2_ reduction [4].

This review compares the insulation properties and the carbon footprint of current insulation materials based on polystyrene, mineral wool, and foamed silicates with a new class of sustainable insulation materials made from natural-fiber-reinforced geopolymer foams.

## 2. Geopolymer-Based Insulation Material

Geopolymers are produced through a polymerization reaction and consequent formation of Si-O-Al bonds between aluminosilicate and alkali polysilicates, e.g., sodium silicate [4,5,6,7]. Commonly, fly ash (FA), ground granulated blast-furnace slag (GGBFS), and metakaolin (MK) are used as precursor materials for aluminosilicate due to their high aluminum (Al) and silicon (Si) content. Slag and fly ash are preferred due to their wide availability and low emissions.

Fly ash is one of the major solid by-products generated by coal combustion for power generation. At present, most fly ash generated all over the world is still dumped as waste material without any beneficial use [8]. Only a small proportion of fly ash is used in applications such as cement and concrete fabrication, mineral wool production, and road construction. Furthermore, fly ash can be used to capture and store atmospheric CO_2_ through mineral carbonation.

To improve the insulating properties of insulation materials, its porosity, thermal conductivity, and mechanical behavior are adjusted through changes in geopolymer composition and synthesis conditions. Low thermal conductivity in particular is one of the essential properties of thermal insulating materials [9,10].

## 3. GWP of Materials Used for the Production of Geopolymer-Based Insulation Material

Geopolymer-based insulation materials are based on sodium silicates and aluminosilicate raw materials, as well as activators, fibers, and fillers. Sources of aluminosilicate could be very different. A very popular material in all regions is fly ash. The binder components of sodium silicates, fly ash, and natural or artificial fibers have a huge impact on GWP. Therefore, we focus on these raw materials. 

### 3.1. GWP of Sodium Silicate

Several GWPs for sodium silicate can be found in the literature. In databases, such as Ecoinvent or SimaPro, different GWPs for sodium silicate are available. Therefore, an overview of the different GWPs for sodium silicate is given in the following chapter and Table 1.

Habert et al. reported a GWP of 1.14 kg CO_2_-eq. kg^−1^ for a sodium silicate solution (37% solid) and 1.76 kg CO_2_-eq. kg^−1^ for spray powder [11]. Turner and Collins calculated 1.22 kg CO_2_-eq. kg^−1^ for emission generated during manufacturing, adding 30% more for transportation, resulting in a total emission of 1.514 kg CO_2_-eq. kg^−1^ sodium silicate (100%). Reasons for the high power consumption include sand sourcing, the Solvay process for producing sodium carbonate, and the high-temperature melting process in which soda ash and sand are mixed [12]. Both Habert et al. (2011) and Turner and Collins (2013) derived their values from Fawer et al. (1999) [13]. Heath et al. (2014) published a GWP value of 0.445 kg CO_2_-eq. for 37% sodium silicate solution taken from the Ecoinvent database [14]. According to Teh et al. (2017), the CO_2_ emissions of 100% sodium silicate are 0.91 kg CO_2_-eq. This value is derived from the Ecoinvent database (v3.2). Furthermore, Teh et al. identified sodium silicate as the main contributor to the overall GWP of geopolymer concrete [4]. A 37% solution was used by Ouellet-Plamondon et al. (2015) with a GWP of 1.14 kg CO_2_-eq. As most manufacturers do not disclose information about their manufacturing processes, it is difficult to obtain data on the emission factors of sodium silicate. Nevertheless, Ouellet-Plamondon et al. identified sodium silicate as the largest contributor to total CO_2_ emissions. The team used values from the Ecoinvent database (v2) and GaBi [11]. The database of the SimaPro7.1 program was used by Mellado et al. (2014) as a source for CO_2_ emission values of commercial sodium silicate. A value of 1.2 kg CO_2_-eq. was calculated [15]. Font et al. (2020) also used the SimaPro database. The sodium silicate with 8% Na_2_O and 28% SiO_2_ was considered in the calculations with an emission value of 1.213 kg CO_2_-eq. by Font [16]. Mastali et al. (2020) stated a GWP of 1.640 kg CO_2_-eq. m^−3^ for sodium silicate with a SiO_2_/Na_2_O ratio of 2.5 according to the Ecoinvent database [17]. Sandanayake et al. (2018) discovered an emission value of 0.78 kg CO_2_-eq. in cooperation with manufacturers [8]. Abbas et al. (2020) used a 40% liquid sodium silicate solution with a SiO_2_/Na_2_O weight ratio of 3:1. After consultation with the Ecoinvent database (v3), SimaPro v8, and other publications, a value of 84.4 kg CO_2_-eq. was considered for sodium silicate [18]. Cristelo et al. (2015) also derived their GWP of 1.096 kg CO_2_-eq. for sodium silicate (SiO_2_/Na_2_O ratio of 2, Na_2_O concentration of 13%) from the Ecoinvent database (v2.2) [19]. Habert and Ouellet-Plamondon differentiated between the emission values for the production in Europe and globally. According to the Ecoinvent database, a 37% sodium silicate solution without water has a GWP of 1.08 kg CO_2_-eq. in Europe and 1.06 kg CO_2_-eq. globally [20]. Sodium metasilicate pentahydrate with a GWP of 1.24 kg CO_2_-eq. was used by Coppola et al. (2018) [21]. Nguyen et al. (2018) derived their value of 0.671 kg CO_2_ for an alkali-activated cement with a mixture of SiO_2_/Na_2_O at a ratio of 2:1. This value was based on calculations from Fawer et al. (1999) [13]. A GWP 100, the GWP over 100 years, of 0.081 kg CO_2_-eq. for sodium silicate (GaBi v6) was used by the team of Manso et al. (2018) in their LCA study [1]. 

According to Alghamdi et al. (2018), the GWP of sodium silicate varies between 1.0 and 1.5 kg CO_2_-eq. kg^−1^ depending on the solids content in the solution. A 36% sodium silicate solution with a SiO_2_/Na_2_O ratio of 3.3 has a GWP of 0.55 kg CO_2_-eq. kg^−1^ [22]. Scrivener et al. also considered a CO_2_ emission value depending on the composition of sodium silicate. Based on the Getting the Numbers Right (GNR) database, the GWP ranges from 0.9 to 1.8 kg CO_2_-eq. For example, a solution with a concentration of 55% has a GWP of 1.1 kg CO_2_-eq. [20]. Rivera et al. (2020) used sodium silicate with 30.18% SiO_2_ and 14.57% Na2O and a GWP 100 of 0.823 kg CO_2_-eq. (Ecoinvent database v3.6) [23]. Sodium metasilicate with a GWP of 0.361 kg CO_2_-eq. g^−1^ was used by Naqi and Jang (2019) [24]. Abdollahnejad et al. (2015) also worked with the Ecoinvent database and chose a value of 1.76 kg CO_2_-eq. for their calculations [25]. Sodium metasilicate pentahydrate used by Coffetti (2019) was noted in the Ecoinvent database (v3) with a GWP of 1.24 kg CO_2_-eq. [26]. Robayo-Salazar et al. (2018) used a GWP 100 of 0.7925 kg CO_2_-eq. in their calculations for commercial sodium silicate, based on the Ecoinvent (v2) database [27]. 

Additionally, they also considered CO_2_ emissions during transportation of the raw material. According to Robayo-Salazar et al., sodium silicate and sodium hydroxide are the main contributors to the total GWP of geopolymer concrete. Alternatives to sodium silicate were investigated, especially rice husk ash, which seems to be a promising substitute. Mellado et al. (2014) also recommended further research for sodium silicate alternatives [15]. In another paper by Robayo-Salazar et al. (2017), a GWP of 0.926 kg CO_2_-eq. was used. This value was taken from the updated version of the Ecoinvent database (v3.2) [28]. Hornich et al. (2015) used the GEMIS database and derived a value of 0.425 kg CO_2_-eq. for 80% sodium silicate solution. ProBas, a database for environmental data by the German environmental federal agency, published two values for sodium silicate: 0.36 kg CO_2_-eq. was attributed to direct emission and 1.9 kg CO_2_-eq. to the processing of sodium silicate [27].

Although most research teams used sodium silicate from local production plants, the majority of emission values were based on the findings of Fawer et al. (1999). The data provided by the Ecoinvent database (v3) were also based on the research of Fawer et al. (1999) with corrections from Davidovits (2015) [5]. This is due to the confidentiality of the producers, as they do not disclose information on the production process. Even if there are local differences in emission values, they may not be detected as one has to rely on the outdated data available.
polymers-14-03191-t001_Table 1Table 1GWP Data of Sodium Silicate—Literature Values and Calculated Values (*) for Comparison of Different Weight Concentrations (36% and 100%) Sorted by Country of Origin.Country of OriginConc. Sodium SilicateLiterature Value[wt%]GWP[kg CO_2_-eq. kg^−1^]GWP Conc. 36 wt% [kg CO_2_-eq. kg^−1^] *GWP Conc. 100 wt% [kg CO_2_-eq. kg^−1^] *SiO_2_ [%]Na_2_O [%]Ref.**Australia**






Turner and Collins (2013)44.11.5141.236 *3.433 *29.414.7[14]Teh et al. (2017)36
0.33 *0.9129.414.7[4] Sandanayake et al. (2018)44.10.780.63 *1.77 *29.414.7[8]**Colombia**






Rivera et al. (2020)44.750.8230.662 *1.84 *30.1814.57[23]Robayo-Salazar et al. (2018)44.010.79250.6483 *1.801 *32.0911.92[27] Robayo-Salazar et al. (2017)44.010.9260.758 *2.10 *32.0911.92[28]**Finland**






Mastali et al. (2020)360.590.590 *1.64071.5 *28.5 *[17]**Germany**






ProBas (2005)1000.737
0.737n.a.n.a.[29]**Italy**






Coppola et al. (2018)360.450.45 *1.2475 *25 *[21]Coffetti et al. (2018)361.24
1.241.92 *4[21]**Portugal**






Cristelo et al. (2015)391.0961.012 *2.810 *2613[19]Abdollahnejad et al. (2017)1001.76
1.76n.a.n.a.[25]**Spain**






Mellado et al. (2014)361.21.20 *3.3 *288[15]Font et al. (2020)361.2131.213 *3.369 *288[16]**South Korea**






Naqi and Jang (2019)1003.61
3.61n.a.n.a.[24]**Switzerland**






Ouellet-Plamondon et al. (2015)36
1.143.08 *28.4 *8.6 *[11]Ouellet-Plamondon et al. (2015)36
0.63 *1.7666.7 *33.3 *[11]Habert and Ouellet-Plamondon (2016)36
1.08 (Europe)1.06 (globally)2.92 * (Europe)2.86 * (globally)n.a.n.a.[30] **United Kingdom**






Scrivener et al. (2018)551.10.72 *2 *n.a.n.a.[31]**USA**






Nguyen et al. (2018)36
0.241 *0.67166.7 *33.3 *[32]Alghamdi et al. (2018)360.550.55 *1.53 *27.7 *8.3 *[22]**Western Europe**






Heath et al. (2014)

0.4451.203 *28.4 *8.6 *[14]Fawer et al. (1999)48288
1.066 *32 *16 *[13]* calculated based on original concentration to compare the different concentrations.


### 3.2. Miscanthus Fibers 

The C4 grass Miscanthus x giganteus, commonly known as elephant grass, is increasingly used in geopolymer production [33,34,35]. The 12-foot-tall plant has suitable thermal insulation properties. Cultivation of Miscanthus is sustainable, as the plant can be grown locally in Europe, needs low amounts of fertilizer, grows on barren soil, is perennial, and absorbs large amounts of CO_2_ [36,37]. Miscanthus can be used in binder systems of cement, lime, and pozzolanic materials (zeolite and fly ash), and it improves thermal insulation; however, mechanical properties can be affected [38].

Miscanthus carries the most leaves in spring and loses a significant portion of them during winter. Harvesting in spring gains the highest possible amount of material for the production of insulation materials [39]. The yearly mean harvestable yields of about 20 tons dry mass ha^−1^ and 20–26 tons ha^−1^ were published by [4] and [37], respectively. Depending on the field of application, the growing conditions, and genotype, the CO_2_ consumption of Miscanthus varies in a wide range. The sequestration of CO_2_ by Miscanthus ranges from 3 to 40 metric tons CO_2_-eq. ha^−1^ depending on geographic location, water availability, and soil quality [20,40,41]. Using a consumption of 20 metric tonnes CO_2_-eq. ha^−1^ and a mean harvestable yield of about 20 metric tonnes dry mass, the mean consumption of GHG for 1 kg dry Miscanthus can be calculated as 1 kg CO_2_-eq. The consumption of CO_2_ by Miscanthus raw material results in a vast improvement of the GWP of the resulting products. 

### 3.3. Fly Ash

Researchers chose different approaches when evaluating the GWP of fly ash; Table 2 gives an overview of the different GWP values used for fly ash. Researchers such as Bajpai et al. (2020) [41], Heath et al. (2014) [14], Assi et al. (2020) [42], Chowdhury et al. (2010) [43], and Ricciotti et al. (2019) [44] attributed no CO_2_ emissions to fly ash. As fly ash is a by-product generated in coal-fired power plants, its production does not contribute to the emissions of the original process. 

Fly ash must undergo specific treatments (grinding, drying, and stock) before it can be used in concrete. Therefore, Habert et al. (2011) considered a GWP of fly ash according to the Ecoinvent database of 5.26 × 10^−3^ kg CO_2_-eq. kg^−1^ [45]. 

Chen et al. (2019) also considered these secondary treatments, resulting in a GWP of 15.894 kg CO_2_-eq. Furthermore, they considered mass allocation and economic allocation with respective emission values of 328.467 kg CO_2_-eq. [46]. Based on information from the LCI database of the Japanese Society of Civil Engineering (JSCE), Yang et al. (2015) reported a GWP of 0.0196 kg CO_2_-eq. kg^−1^ for fly ash [20]. Bajpai et al. (2021) considered a GWP of high-CaO fly ash according to the South Korean LCI database of 0.0101 kg CO_2_-eq. kg^−1^ [41]. 

Gunasekara et al. (2018) considered a GWP of 0.0032 kg CO_2_-eq. kg^−1^ for fly ash. The transport emissions were included with 0.0018 kg CO_2_-eq. kg^−1^ [8,44,47]. Ohenoja et al. (2020) investigated the sequestration properties of fly ash. It was discovered that 6.7 metric tons of fly ash can capture one metric ton of CO_2_ [8,45]. Nguyen et al. (2018) performed calculations with a GWP of 0.006 kg CO_2_-eq. kg^−1^ attributed to the grinding and collecting of fly ash [32]. These processes were also considered by Flower and Sanjayan (2007), including transportation. Thus, a GWP of 0.027 kg CO_2_-eq. kg^−1^ was adopted for fly ash [48].

Teixeira et al. (2016) based their calculations on a GWP 100 value for coal fly ash of 1.01 × 10^−2^ kg CO_2_-eq. from the Ecoinvent database (v2.2) [49]. An environmental product declaration (EPD) for fly ash by the Danish Technological Institute (2013) states a total GWP of 3.92 kg CO_2_-eq. This value includes loading at the power plant, storage, and transport [50].

Kurda et al. (2018) applied a GWP value of 0.004 kg CO_2_-eq. in their Native LCA method, in which the data considered originated from only the country of interest [51]. An emission value of 0.0132 kg CO_2_-kg^−1^ for fly ash from the EASETECH waste management software was used by Naroznova et al. (2016) [52]. Hossain et al. (2017) used several LCI databases and selected a GWP of 0.006 kg CO_2_-eq. for their calculations [53].

The cut-off approach of Arrigoni et al. (2020) considered only transportation and processing, resulting in emissions of 24 g CO_2_-eq. kg^−1^. If, on the other hand, the substitution approach was chosen, the avoided disposal of fly ash resulted in 62 g CO_2_-eq. kg^−1^ [54]. Balaguera et al. (2019) also chose the cut-off rule according to ISO 14044:2006 and considered only transportation with a GWP 100 of 2.89 × 10^−1^ kg CO_2_-eq. [55].polymers-14-03191-t002_Table 2Table 2GWP Data of Fly Ash Sorted by Country of Origin.Country of OriginGWP[kg CO_2_-eq.]Allocation of CO_2_-eq. EmissionsRef.**Australia**


Flower and Sanjayan (2007)27processing, transport[48]Gunasekara et al. (2021)0.0032material extraction to production,transport[56]**Canada**


Arrigoni et al. (2020)0.024 (cut-off approach)−0.062 (substitution approach)transport;avoided disposal[54]**Colombia**


Balaguera et al. (2019)2.89 × 10^−1^transport[55]**Denmark**


Naroznova et al. (2016)0.0132production and combustion,landfill, electricity, fuel[52]**Finland**


Ohenoja et al. (2020)−0.15 *sequestration[47]**France**


Habert et al. (2011)5.26 × 10^−3^processing[45] **Hong Kong**


Hossain et al. (2016)0.006collection, processing, transport[53]**Portugal**


Kurda et al. (2018)0.004economic allocation (combustion, extraction, transport)[51]Teixeira et al. (2016)1.01 × 10^−2^classification, combustion, extraction, transport[49]**South Korea**


Lee at al. (2021)1.73 × 10^−3^economic allocation (combustion, extraction, transport)[57]Yang et al. (2015)0.0196processing, storage[58]**USA**


Chen et al. (2019)15.8942584.743 (mass allocation)328.467 (economic allocation)processing, transport[46] Nguyen et al. (2018)0.006collection, processing[32]* calculated based on values in the literature.


## 4. Insulation Materials for Wall Construction

According to European regulations, building materials have to be declared regarding their impact on the environment with environmental product declarations (EPD). In Germany, about 55% of thermal insulations are based on mineral wool and 40% on polystyrene [59]. Polystyrene is the preferred insulation material in the segment of thermal insulation composite systems (ETICS) with approx. 80% market share, whereas mineral wool has a share of 20%. Disposal or recycling of those materials is associated with problems. Insulation materials made of polystyrene contain flame retardants, usually hexabromocyclododecane (HBCD), a substance which fulfills the criteria for PBT substances according to the European chemicals regulation REACH. It is persistent (permanently remaining in the environment), bioaccumulating (occurring in organisms), and toxic (poisonous for humans, ecosystems, and organisms). Statistical data of the Federal Institute for Research on Building, Urban Affairs, and Spatial Development provides information regarding only the total amount of annually accumulating insulation materials (85,000 tons), not differentiation of individual materials. The amount of expanded polystyrene (EPS) construction waste is about 42,000 tons per year [59]. The use of renewable thermal insulation products achieves excellent insulation properties and reduces the GWP. Furthermore, the reduced impact regarding density, disposal, and recycling is a major advantage of those products. Most of the natural fibers have low environmental impacts regarding transformation and extraction processes (Table 3). Nevertheless, they are not flame resistant at all. The combination of renewable plant fiber materials and flame resistant geopolymer mineral binders with low GWP, such as sodium silicate, provides an effective way to improve fire resistance and the environmental impact.

## 5. Calculation of GWP of Geopolymer-Based Insulation Material with Miscanthus Fibers

Porous geopolymers containing Miscanthus fibers are formed through the polymerization of fly ash (FA) with sodium silicate. The reaction is supported by a foaming process with sodium dodecyl sulfate [61]. The purpose of this study is to evaluate the CO_2_ emission of different formulations of foamed geopolymers via life cycle assessment (LCA). For comparison of different insulation materials with regard to their thermal conductivity, the thickness of the insulation layer must be considered. For normalization of this variable, the heat transfer coefficient, also known as the U-value, is used. 

The U-value of one layer is calculated by dividing the thermal conductivity of that layer by its thickness. The thermal resistance, R, of the layer is calculated by dividing the thickness of the layer by its thermal conductivity. Current requirements for conventional new buildings in Germany prescribe a heat transfer coefficient of U = 0.24 W m^−^² K^−1^ for external or internal wall insulation (Figure 1 (right)).

The general formula for calculating the U-value is U = 1/Rt, where U = thermal transmittance (W/m²·K); Rt = total thermal resistance of the element composed of layers (m²·K/W), obtained according to Rt = Rsi + R1 + R2 + R3 + … + Rn + Rse; Rsi = interior surface thermal resistance (according to the norm by climatic zone); Rse = exterior surface thermal resistance (according to the norm by climatic zone); and R1, R2, R3, …, Rn = thermal resistance of each layer. R is calculated as R = D/λ, where D = material thickness (m) and λ = thermal conductivity of the material (W/K·m) (according to each material).

The U-value of two or more layers is calculated by adding the thermal resistances of each separate layer that make up the element to the thermal resistance of the inside and outside surface. For example, a brick wall with U = 2 W m^−^² K^−1^ is assumed as the first layer combined with an insulation layer (Figure 1 (left)). Depending on the thermal conductivity of the insulation material, it is possible to calculate the thickness of the insulation layer necessary to achieve U = 0.24 W m^−2^ K^−1^ and the GWP for the amount of material needed.

LCA was performed with four formulations with different fiber contents to evaluate the environmental influences associated with these new insulation materials. In the GWP calculations, Class F fly ash was considered with a CO_2_ equivalent of 0.006 kg CO_2_-eq. according to Nguyen et al. [18] and sodium silicate with a CO_2_ equivalent of 0.424 kg CO_2_-eq. (solid content of 36%; 30.28% SiO_2_, 14.57% Na_2_O, and 54.79% H_2_O) according to Coppola [18]. The GWP consumption of Miscanthus was included in the calculations as 1 kg CO_2_-eq. kg^−1^. Global warming potential (GWP) of the main binder component, sodium silicate, is published in a range from 0.3 to 3.3 kg CO_2_-eq. kg^−1^. The CO_2_ emissions of sodium silicate vary depending on the local production, transportation, and respective energy consumption during these steps. The calculation in GF04* was performed with a value of 1.25 kg CO_2_-eq. kg^−1^ to consider the production in regions with higher GWP specifications of sodium silicate. Furthermore, the production impact was considered with a CO_2_ equivalent of 0.05 kg CO_2_-eq. Table 3 gives a summary of the GWP calculations for formulations GF01 to GF04 with increasing Miscanthus. 

## 6. Conclusions

This study evaluated the environmental impact of sustainable insulation materials based on Miscanthus x giganteus fibers, sodium silicate, foaming agents, and alkali-activated fly ash binder. The GWP of the foamed insulation boards was estimated. From the data evaluated in this study, the following conclusions can be drawn:The GWP of the sodium silicate strongly depends on the production, transportation, and energy consumption and is published in a range from 0.3 kg to 3.3 kg CO_2_-eq. kg^−1^.The consumption of greenhouse gas by Miscanthus × giganteus varies in a range from 3 to 40 metric tons CO_2_-eq. ha^−1^. Using a mean consumption of 20 tons CO_2_-eq. ha^−1^ and a mean harvestable yield of about 20 tons dry mass, the mean consumption of greenhouse gas for 1 kg Miscanthus was calculated as 1 kg CO_2_-eq.The GWP of all formulations is lower compared to systems based on mineral wool and polystyrene (EPS). The formulation with the lowest impact has the highest fiber content of 40%.According to the current regulations for thermal insulation materials, the resulting GWP of geopolymer-based insulation materials with Miscanthus fibers is 10% or lower compared to conventional systems.

## Figures and Tables

**Figure 1 polymers-14-03191-f001:**
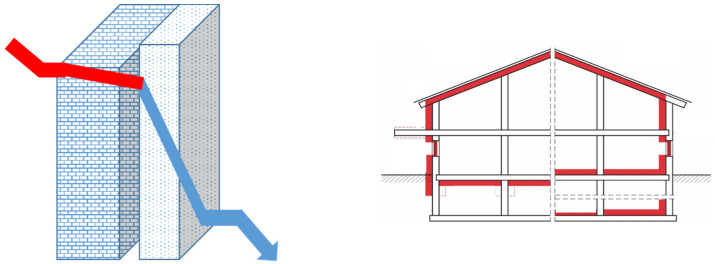
Heat flow through a brick wall combined with an insulation layer (**left**) and possible external and internal positions of insulation material in house construction alongside walls, roof, and flooring (**right**).

**Table 3 polymers-14-03191-t003:** Specification of Thermal Conductivity of Popular Insulation Materials and Calculation of GWP for 1 m² Wall Insulation of U = 0.24 W m^−2^ K^−1^.

Properties	Density	Thermal Conductivity	GWP for 1 m³ Insulation Material	Thickness of Insulation	GWP for 1 m² Wall Insulation	Ref.
	[kg m^−3^]	[W m^−1^ K^−1^]	[kg CO_2_ m^−^³]	[cm]	[kg CO_2_-eq.]	
**Polystyrene EPS 035**	30	0.035	75	15	389	[59]
**Hemp fiber**	50	0.04	−96	17	−16	[59]
**Miscanthus fiber**	40	0.035	−82	15	−12	[59]
**Mineral wool**	80	0.035	37	15	432	[60]
**Foamed concrete**	250	0.07	167	29	12,177	[59]
**Foamed concrete**	400	0.1	167	42	27,833	[59]
**Wood (OSB)**	650	0.13	−500	54	−35	[59]

## Data Availability

Not applicable.

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
