# Peer review of "Minimizing the Global Warming Potential with Geopolymer-Based Insulation Material with Miscanthus Fiber"

_polymers, 2022, doi:10.3390/polym14153191_

Round 1

Reviewer 1 Report

The review by this Author treats a very specific topic because it does not deal with a class o geopolymers but with those employing Miscanthus fiber. In particular, the aim of the Author was to evaluate, by using LCA data, the reducing effects on the GWP of these natural fiber-based Geopolymers when used as insulation material.  The architecture of the manuscript is unusual for a review format, and in addition, some sentences in the Introduction can generate confusion for the reader, because sometimes (during the reading) it seems that the manuscript is a research article than a review (please correct). The introduction must be supported by references in some aspects (see attached report), the tables need care and the whole manuscript needs a strong revision to eliminate many typos and mistakes (see attached report). Focusing on the core of the review, e.g. the use of Miscanthus fibers, I think that sustainability must consider also the soil availability and not only the fact it requires a low amount of fertilizer or carbon dioxide absorption.

Author Response

Thank you for the comments and recommendations. 

The introduction was changes according attached report.

The tables and manuscript were revisied to eliminate typos and mistakes.

Hints regarding use of Miscanthus fibers were added.

Reviewer 2 Report

- Line 62. Section 3 "GWP of materials used for the production of Geopolymer based insulation material - literature value". Please explain why this section contains only 3 components used for the synthesis of Geopolymer based insulation materials: sodium silicate, fly ash and Miscanthus fiber. Normally, for the synthesis of geopolymers a wide and varied range of aluminosilicate raw materials, activators and fillers are used.

- Line 64. In Section 3.1, replace "silicat" with "silicate"

- Line 152. Section 3.2 should explain how Miscanthus fibers affect GWP reduction when Geopolymer based insulation material production

 - Line 169. Please explain the abbreviation GHG

- Line 294. In the text, the authors refer to Table 4. However, Table 4 is missing in the article.

Author Response

Thank you for the comments and suggestions. Corrections have been added.

Round 2

Reviewer 1 Report

The manuscript in the revised version can be accepted for publication

Reviewer 2 Report

Manuscript can be accepted in present form